# Flexible Optimal Control of the CFBB Combustion System Based on ESKF and MPC

**DOI:** 10.3390/s25041262

**Published:** 2025-02-19

**Authors:** Lei Han, Lingmei Wang, Enlong Meng, Yushan Liu, Shaoping Yin

**Affiliations:** 1School of Computer and Information Technology, Shanxi University, Taiyuan 030006, China; 13834163454@163.com; 2School of Automation and Software, Shanxi University, Taiyuan 030006, China; 13546468676@163.com (E.M.); 18234004469@163.com (Y.L.); yin297@126.com (S.Y.)

**Keywords:** extended state Kalman filter, model predictive control, circulating fluidized bed boiler, combustion system

## Abstract

In order to deeply absorb the power generation of new energy, coal-fired circulating fluidized bed units are widely required to participate in power grid dispatching. However, the combustion system of the units faces problems such as decreased control performance, strong coupling of controlled signals, and multiple interferences in measurement signals during flexible operation. To this end, this paper proposes a model predictive control (MPC) scheme based on the extended state Kalman filter (ESKF). This scheme optimizes the MPC control framework. The ESKF is used to filter the collected output signals and jointly estimate the state and disturbance quantities in real time, thus promptly establishing a prediction model that reflects the true state of the system. Subsequently, taking the minimum output signal deviation of the main steam pressure and bed temperature and the control signal increment as objectives, a coordinated receding horizon optimization is carried out to obtain the optimal control signal of the control system within each control cycle. Tracking, anti-interference, and robustness experiments were designed to compare the control effects of ESKF-MPC, ID-PI, ID-LADRC, and MPC. The research results show that, when the system parameters had a ±30% perturbation, the adjustment time range of the main steam pressure and bed temperature loops of this method were 770~1600 s and 460~1100 s, respectively, and the ITAE indicator ranges were 0.615 × 10^5^~1.74 × 10^5^ and 3.9 × 10^6^~6.75 × 10^6^, respectively. The overall indicator values were smaller and more concentrated, and the robustness was stronger. In addition, the test results of the actual continuous variable condition process of the unit show that, compared with the PI strategy, after adopting the ESKF-MPC strategy, the overshoot of the main steam pressure loop of the combustion system was small, and the output signal was stable; the fluctuation range of the bed temperature loop was small, and the signal tracking was smooth; the overall control performance of the system was significantly improved.

## 1. Introduction

In recent years, with the prominence of the global energy crisis and environmental protection issues, new energy has witnessed rapid development, and the complexity of the global energy system has been further intensified. During the process of integrating new energy power generation, the demand for flexibility in power systems has increased significantly. As stable and controllable thermal power-generating units, they have taken on more flexible regulation tasks to ensure the safety and stability of the power system. Circulating fluidized bed units have become one of the main forces for the peak load regulation of the power grid due to their wide load regulation capabilities. However, the complex dynamic response process and significant dynamic response delay on the boiler side severely restrict the rapid response and tracking performance of the combustion control system. Therefore, against the technical background of ensuring that units can efficiently adapt to power grid peak-shaving, there is an urgent need to optimize the complex, coupled, and time-varying circulating fluidized bed combustion control system, so as to effectively improve the control performance of the entire system during the flexible operation of the unit.

The boiler combustion control system involves numerous control variables and controlled variables, which are mutually coupled and influence each other. The main steam pressure and the bed temperature, as the two most important controlled variables in the combustion control system, exhibit a strong correlation. When one of them participates in regulation, the other will inevitably be affected. The nonlinear coupling relationship between them becomes even more complex as the unit operates flexibly. Moreover, the main steam pressure and the bed temperature have different dynamic response characteristics. Specifically, the main steam pressure responds more quickly to regulatory actions, while the bed temperature responds more slowly. The reason for this lies in the fact that the heat storage bed material in the furnace has a relatively large heat capacity, and it takes a certain process for the bed temperature to rise or fall. Under the existing PI control strategy, the delay characteristics of the bed temperature during the flexible operation of the unit are constantly changing, which increases the difficulty of stable bed temperature operation. Affected by coupling, the main steam pressure is also prone to fluctuations and difficult to balance. To address this, scholars at home and abroad have carried out a series of studies on the complex combustion system of circulating fluidized beds, adopting many advanced and new control algorithms, including the fuzzy control strategy [1,2,3,4], active disturbance rejection control (ADRC) strategy [5,6,7,8,9,10,11], and MPC strategy [12,13,14,15,16,17,18,19].

In the fuzzy control strategy, the idea of “fuzzy” is used to eliminate the output signal deviation caused by the internal and external disturbance of complex nonlinear controlled objects. This control strategy can be used alone or combined with the traditional PID controller. For example, the authors of [4] propose an intelligent coordinated control strategy based on the joint action of fuzzy feedforward and fuzzy PID feedback to solve the problems of strong coupling, nonlinearity, and instability of the combustion system of the CFB unit, which effectively improves the regulating performance of the control system. Considering the complexity and high cost of establishing fuzzy rules in fuzzy controllers, it is difficult to apply them on a large scale in industrial processes. ADRC, as a new algorithm integrating classical control theory and modern control theory, compensates uncertain objects to integral series by means of an extended state observer designed to improve the robustness and adaptability of the initial setting controller. The standard active disturbance rejection controller is a nonlinear controller, and its parameter tuning is more complicated, but Dr. Gao Zhiqiang [20] converted it to linear ADRC, which greatly promoted its application in engineering practice. In the active disturbance rejection control algorithm, the high-order characteristics, uncertainties, and unmodeled dynamics of CFBB combustion system objects can be regarded as disturbances, which are estimated and actively compensated by an ESO, and the control system shows certain robustness and disturbance immunity. However, the control function of the algorithm is limited due to the large time-delay term contained in the system. The main performance is that the output of the system entering the extended state observer is not synchronized with the control quantity, and the time deviation at the interval is the time-delay quantity. In [21,22,23], a MADRC strategy was proposed, which was characterized by transforming the compensation link of the control quantity into an inertia link. Reference [21] proposed a MADRC control scheme for the coupled and high-order controlled system of an FBC boiler. The simulation results show that, compared with other proportional–integral controllers and ADRC, MADRC had the minimum overshoot and the shortest stabilization time and exhibited the strongest ability to suppress interferences such as coal quality variations. Reference [22] addressed the problem of a sluggish disturbance response in the superheated steam temperature control system of a 300 MW CFB. It proposed an improved cascade control strategy based on an active disturbance rejection controller. Field applications indicated that this strategy could significantly reduce the temperature deviation under both large and small load changes. In view of the strong nonlinearity and large time delay of the CFB control system, Reference [23] constructed a MADRC strategy based on gain scheduling. Through comparative simulations, the advantages of this method in terms of tracking performance, anti-interference, suppression of combustion quality fluctuations, and uncertain heat transfer coefficients were verified, and the proposed strategy was practically applied to the main steam pressure system. References [24,25,26] proposed a control strategy using Smith’s predictive feedforward compensation. Wang Y.S. et al. [24] proposed a method combining a Smith predictor with linear active disturbance rejection control for a first-order inertial system with a large time delay. It was verified that this method could achieve an ideal control effect. In order to improve the performance of first-order time-delay systems, G.D. Chen et al. [25] proposed a new Smith predictor combined with LADRC, which solves the problem of asynchronous time scales of the two input signals of LESO. B. Zhang et al. [26] developed a control strategy that combined GADRC and SP for time-delay systems, fully inheriting the advantages of both SP-ADRC and GADRC. The authors of [27,28] proposed another improvement method and designed a new lag time weakening structure. Liu Y.C. et al. [27] proposed using ADRC along with two modified structures applying the Smith predictor and time-delay estimation, respectively, to conduct a study on the trade-off between the performance and robustness of a class of time-delay systems. Li X.Y. et al. [28] proposed a time-delay weakening and active disturbance rejection control method for uncertain systems with variable time delay, variable parameters, and disturbances. This method comprehensively applied feedforward control, feedback control, and active disturbance rejection compensation control. Theoretical proofs and simulation results show that the proposed method was effective and had good control performance. In addition, ADRC is essentially a strategy used to deal with single-loop control. In the face of multi-input, multi-output coupling control loops, relevant measures should be taken to eliminate coupling effects to maintain ADRC’s efficient performance. The authors of [29,30] developed a practical multi-variable control method for a TITO system, which was a combination of inverse decoupling and decentralized active disturbance rejection, and conducted a full experimental verification.

Although ADRC can effectively address nonlinearities and large inertia in control systems, the standard ADRC algorithm often falls short in handling issues like coupling and significant time lags. To address these challenges, an improved algorithm is typically required. Predictive control naturally adapts to these issues through its strategies of multi-step prediction, rolling optimization, and feedback correction. This approach facilitates global optimization of the adjustment process while balancing rapidity, stability, and accuracy. While MPC can achieve optimal control in nonlinear coupled systems, its accuracy is highly dependent on the precision of the model. If there is a significant deviation between the model parameters and the actual system, the control system may fail to compute the correct control variables when evaluating the cost function, potentially leading to degraded system performance. How to ensure the accuracy of the reference model is a very important problem. The authors of [31,32] proposed the MPC control scheme based on LPV model modeling in view of the variation characteristics of CFB boiler’s wide load output, and the results showed that this method produced a good control effect in the frequently changing range of unit loads compared with the fixed linear parameter model. However, the authors of [33] proposed using an observer to estimate the system’s state and disturbance to improve the MPC algorithm and improve its performance.

Considering the harsh working environment of the CFBB combustion system, multiple sources of interference signals, and the complex time-delay coupling characteristics of the system itself, it is very difficult for the traditional PI control scheme to guarantee the control quality during the flexible operation of the system. In this paper, MPC, which is applicable to multi-variable control systems, is taken as the framework for rolling optimization and adjustment. Drawing on the idea of active disturbance rejection control, the ESKF [34] is used to estimate and compensate for the system uncertainties caused by interference measurement signals, model coupling nonlinearities, and other factors. The simulation results show that, under the new control scheme, the overall response speed and stability of the combustion system have been significantly improved. The main contributions of this paper are as follows:(1)For the first time, a method of enhancing the robustness of MPC control using the ESKF algorithm is applied to the combustion system of a circulating fluidized bed boiler with complex coupling characteristics, in order to effectively improve the overall performance of the control system during the flexible operation process. This solution helps to reduce the frequent fluctuations of the main steam pressure and the bed temperature, ensuring the efficiency of the operation process;(2)The extended state Kalman filter is used to ensure the accuracy and stability of the collected signals. This is especially true for the bed temperature signals in the high-temperature furnace during the combustion fluctuation process, providing stable and accurate signals for feedback control. Based on this control system, unnecessary frequent actions and wear of equipment can be reduced, equipment failures can be decreased, and the service life of equipment can be extended;(3)Combining the ESKF to conduct online estimation and compensation of the “external disturbances” of the system significantly reduces the dependence of the MPC control strategy on an accurate model. This improves the adaptability and eases the modeling difficulty during the implementation of the MPC strategy. It is conducive to the widespread application of its online optimization features.(4)During the continuous load-rising and load-falling process of CFB units, the combustion system adopting the ESKF-MPC control strategy exhibits faster tracking ability, better anti-disturbance performance, and enhanced robustness compared to the PI control strategy. Simulation results show that, for a 330 MW CFB unit at the rated capacity, when the load condition fluctuates by up to 50%, the controller still demonstrates reliable robust control performance.

## 2. CFB Boiler Combustion System Description

The combustion system of the CFB boiler represents a highly coupled multi-input, multi-output system, with the bed temperature and main steam pressure being paramount parameters. As a critical state variable, the bed temperature directly reflects the thermal balance within the furnace. Exceeding the optimal desulfurization temperature threshold can lead to an imbalance in the calcium–sulfur ratio, significantly reducing desulfurization efficiency, raising NOx emissions in flue gas, and escalating the cost of pollution control. Conversely, if the bed temperature is too low, it can reduce boiler efficiency, decrease power output, and, in severe cases, lead to fire suppression and shutdown. The main steam pressure parameter highlights the energy balance on the unit’s furnace side, and fluctuations here can negatively impact the stability of other parameters, adding complexity and difficulty to the overall system control adjustment. The regulation of the main steam pressure involves fuel combustion and heat transfer processes, presenting a classic example of a control system with high inertia, requiring strategies that address slow response times and inadequate interference mitigation capabilities [22].

As can be seen from Figure 1, the combustion process of a CFBB furnace exhibits large thermal inertia due to the large amount of heat storage bed material stored in the furnace, while the fluidized combustion mode of residual carbon particles in the furnace increases the difficulty of establishing an accurate mathematical model of the combustion process [35,36,37,38,39]. Although both the primary and secondary wind contain a large amount of air flow, which has an important impact on the combustion in the furnace, the primary wind has the greatest impact on the bed temperature, main steam pressure, flue gas oxygen content, and other parameters, while the secondary wind has the greatest impact on the flue gas oxygen content parameters, and the bed temperature is relatively weak. Therefore, in analyzing the dynamic characteristics of the CFBB combustion system, the simplified dual-input, dual-output model is adopted. The inputs usually refer to the coal feed amount (kg/s) and the primary air volume (Nm^3^/h), and the outputs are the main steam pressure (MPa) and the bed temperature (°C). There are important relationships between the inputs and outputs, which can be described by the high-order constant volume delay function $G(s)=\frac{Ke^{-\tau s}}{{\left(Ts+1\right)}^{n}}$.

## 3. Control Strategy Design

### 3.1. ESKF-MPC Algorithm

The ESKF-MPC algorithm incorporates ESKF into the MPC algorithm, which is essentially a variant of MPC. Standard MPC [40] is a multi-objective rolling optimization algorithm, and its working principle can be summarized in three steps: prediction, optimization, and receding time-domain application. The prediction step involves predicting the future states and outputs of the control system over a specified time horizon. This prediction is based on a non-parametric or parametric model that has been established for the system. In the optimization step, the MPC algorithm calculates the optimal outputs and control inputs for the control system. It does this by minimizing a defined cost function while adhering to specified constraints. In the receding time-domain application step, the MPC algorithm implements the optimal control sequence derived from the previous steps over the current time period. It then repeats the prediction and optimization steps at the next time step, continually updating and applying the optimal control strategy.

Among the three steps, prediction is the most crucial, as the accuracy of the prediction model directly impacts the quality of control. To ensure an accurate prediction model, the optimal estimation function of the ESKF is employed. The ESKF, an enhancement of the Kalman filter, is an optimal recursive algorithm that uses Kalman gain to dynamically adjust the predicted system state, progressively aligning it with the actual state. This paper addresses the issue where quantization errors, measurement errors, and other disturbances in the estimation process of the traditional linear Kalman filter can lead to increased or even divergent system state estimation errors. In summary, the ESKF is first utilized to address unknown disturbances and uncertainties in the control system. Subsequently, the MPC rolling optimization function is applied to further correct any residual errors not fully compensated by the ESKF, thereby effectively minimizing the deviation from the reference trajectory in optimal system control.

Assuming that the state-space model of a certain system is shown in Equation (1), the state estimator of the controlled system is designed by using the extended Kalman filter theory, and the state quantity of the system at every moment is estimated.(1)$$\left\{\begin{array}{l} \dot{X}=AX+BU+EF\\ Y=CX+n \end{array}\right.$$

A Zero order holder is used to discretize the state-space expression of the system, and the perturbation term in the system is extended to a new variable. At this time, the state variable of the discrete expansion model of the system is $Z={\left[\begin{array}{cc} X(k) & F(k) \end{array}\right]}^{T}$, and the model expression reconstructed based on this state variable is as follows:(2)$$\left\{\begin{array}{l} Z(k+1)={\bar{A}}^{\prime }Z(k)+{\bar{B}}^{\prime }U(k)\\ Y(k)={\bar{C}}^{\prime }Z(k) \end{array}\right.$$

${\bar{A}}^{\prime }=\left[\begin{array}{cc} A & E\\ 0 & I \end{array}\right]$,${\bar{B}}^{\prime }=\left[\begin{array}{c} B\\ 0 \end{array}\right]$,${\bar{C}}^{\prime }=\left[\begin{array}{cc} C & 0 \end{array}\right]$.

Based on the above equation, the ESKF observation model is obtained as follows [41]:(3)$$\left\{\begin{array}{l} \hat{Z}(k+1)={\bar{A}}^{\prime }\hat{Z}(k)+{\bar{B}}^{\prime }U(k)+K_{k}(Y(k)-{\bar{C}}^{\prime }\hat{Z}(k))\\ \hat{Y}(k)={\bar{C}}^{\prime }\hat{Z}(k) \end{array}\right.$$

The dynamically adjusted Kalman gain coefficient can be calculated and updated by the following formula:(4)$$K_{k}=\frac{P_{k}^{-}{\bar{C}}^{\prime }}{{\bar{C}}^{\prime }P_{k}^{-}{{\bar{C}}^{\prime }}^{T}+R_{k}}$$

The update of the covariance matrix of the prior estimates is calculated by the following two equations:(5)$$P_{k}=(I-K_{k}{\bar{C}}^{\prime })P_{k}^{-}$$(6)$$P_{k}^{-}={\bar{A}}^{\prime }P_{k-1}{{\bar{A}}^{\prime }}^{T}+Q_{k}$$

**Hypothesis** **1**.*The initial values of the system disturbance* $F\left(k\right)$ *and the estimated error are bounded,* $\underset{k\geq 0}{\sup}\left\|F\left(k\right)\right\|\leq M_{f}$*,*$\left\|Z\left(0\right)-\;\hat{Z}\left(0\right)\right\|\leq \rho_{0}$*, where* $M_{f}$ *and*$\rho_{0}$ *are normal numbers.*

**Lemma** **1.** **[41,42]:**Under the condition of hypothesis 1, if we consider the system model (2) and ESKF observation Equation (3), and assume that the estimation error is $\hat{E}\left(k\right)=Z\left(k\right)-\hat{Z}\left(k\right)$*, then the estimation error at the next time is *
$\hat{E}\left(k+1\right)={\bar{A}}^{\prime }\left(I-K_{k}{\bar{C}}^{\prime }\right)\hat{E}\left(k\right)$
*, and the stability and convergence of the ESKF must meet the following conditions:*
(7)$$\left\{\begin{array}{l} \left\|\hat{E}\left(k\right)\right\|\leq \eta \left\|\hat{E}\left(0\right)\right\|\alpha^{k}\\ \left\|K_{k}\right\|\leq M_{k} \end{array}\right.$$
*where*
0<α<1*,* η *and*
$M_{k}$ *are normal numbers.*

The ESKF algorithm is used to estimate the state of the control system accurately so that the output of the model is consistent with that of the real system. The above process can be regarded as the prediction step in model predictive control, and the principle of the optimization solution step in model predictive control is as follows.

In the process of the optimization solution, the cost function is usually established with the goal of system output deviation and control smoothness. The specific expression is as follows:(8)$$\begin{array}{c} J_{\min}=\sum_{i=1}^{P}{\left(Rs(k+i)-Y(k+i)\right)}^{T}W_{1}\left(Rs(k+i)-Y(k+i)\right)+\\ \;r\sum_{j=1}^{M}\left(\Delta U{\left(k+j-1\right)}^{T}W_{2}\Delta U(k+j-1)\right) \end{array}$$

By calculating the derivative of the cost function with respect to the control increment, the optimal control sequence at the current time is obtained.(9)$$\frac{\partial J}{\partial \Delta U}=-2\Phi^{T}(Rs-FX)+2\Delta U(\Phi^{T}\Phi +R)$$

In the formula, $\Phi =\left[\begin{array}{cccc} \tilde{C}\tilde{B} & 0 & \cdots & 0\\ \tilde{C}\tilde{A}\tilde{B} & \tilde{C}\tilde{B} & \cdots & 0\\ \vdots & \vdots & \ddots & \vdots \\ \tilde{C}{\tilde{A}}^{P-1}\tilde{B} & \tilde{C}{\tilde{A}}^{P-2}\tilde{B} & \cdots & \tilde{C}{\tilde{A}}^{P-M}\tilde{B} \end{array}\right]$ and $F={\left[\begin{array}{cccc} \tilde{C}\tilde{A} & \tilde{C}{\tilde{A}}^{2} & \cdots & \tilde{C}{\tilde{A}}^{P} \end{array}\right]}^{T}$.

It can be seen from Equation (9) that when the minimum value is taken, $\Delta U={\left(\Phi^{T}\Phi +R\right)}^{-1}\Phi^{T}(Rs-FX)$. After obtaining the optimal control increment at each time under constraint conditions, $u^{*}(k)$, the optimal control sequence is calculated by taking the first term.

In the process of solving the minimum value of the cost function, the constraints of the control quantity of the system must be considered.(10)$$u_{\min}\leq u(k+i-1)\leq u_{\max},j=1,2,\cdots ,P$$

The upper and lower limits of the control quantity and control increment can be set according to the real adjustment output characteristics of the controlled system.

The receding time-domain optimization is a cyclic process that can be repeated according to the marking results of the current moment.

The stability proof of this algorithm can be found in reference [43]. The MPC algorithm integrated with the ESKF is summarized in Figure 2.

### 3.2. Controller Design

As CFB units frequently engage in load scheduling operations for the power grid, the uncertainty associated with parameter fluctuation frequencies and disturbance variables in the boiler combustion system model significantly increases. The single-feedback correction control method used by traditional MPC controllers has limited ability to correct system deviations effectively. To enhance the controller’s adaptability, these uncertain disturbances must be estimated and compensated efficiently. Thus, the concept of incorporating an extended state observer (ESO) into the ADRC algorithm is employed. This approach expands the internal and external disturbances of the CFB combustion system into a new variable. Combined with the Kalman filter’s ability to suppress noise and dynamically adjust the observed gain coefficient, the ESKF is established. This filter enables timely optimal estimation of both system states and uncertain disturbances, thereby improving the robust control performance of the MPC controller.

The model predictive control architecture based on the extended state Kalman filter is shown in Figure 3.

In this algorithm, the ESKF is utilized to recursively compute the state variables and total disturbances of the CFBB combustion system. These estimates are then fed into the MPC prediction model to enhance its predictive accuracy. Subsequently, a cost function is employed in the optimization process to determine the control inputs for coal feed and primary air in the combustion system. The Kalman gain coefficient in the ESKF is constantly changing with the change in the system’s state. When the initial system error in the control is large, the Kalman gain coefficient in the ESKF is set to a high value. As the system error gradually reduces, the Kalman gain coefficient decreases accordingly. In summary, this algorithm can be used as feedforward compensation for the MPC system, reducing the impact of disturbances on the performance of the MPC controller. It enables fast and stable control of the combustion system even in the presence of varying interference sources and measurement noise.

## 4. Simulation Results and Analysis

To verify the reliability of the proposed algorithm, a 350 MW circulating fluidized bed unit was used as the research object to verify the algorithm. The selected transfer function model of CFB unit combustion system identification is as follows [44]:(11)$$\left[\begin{array}{l} P(s)\\ T(s) \end{array}\right]=\left[\begin{array}{cc} \frac{2.6e^{-100s}}{{\left(260s+1\right)}^{2}} & \frac{3.3e^{-40s}}{{\left(150s+1\right)}^{2}}\\ \frac{5.6e^{-60s}}{{\left(180s+1\right)}^{2}} & \frac{-11.8e^{-30s}}{{\left(163s+1\right)}^{2}} \end{array}\right]\left[\begin{array}{l} M(s)\\ V(s) \end{array}\right]$$

The relative gain of the current combustion system can be calculated as follows:(12)$$\Lambda =p\times {\left(p^{-1}\right)}^{T}=\left[\begin{array}{cc} 0.6241 & 0.3759\\ 0.3759 & 0.6241 \end{array}\right]$$

From the calculation results of the relative gain, we can see that the gain values of the two circuits are between 0.3 and 0.7, indicating that there is a very serious coupling relationship. Considering that PI and LADRC control algorithms are not suitable for multi-input, multi-output coupled systems, in order to better ensure the efficiency of the algorithm, it is necessary to decouple the two control strategies when selecting them for comparison. In this paper, the CFBB combustion system with two inputs and two outputs is decoupled into two single circuits by the method of inverse decoupling. The control structure of inverse decoupling is shown in Figure 4.

It can be seen from the control structure in Figure 4 that the coupled interactions in the system can be regarded as external disturbances, and the influence of these disturbances can be eliminated by means of the feedforward compensation design, that is, series compensation links based on the original inverse decoupling. Based on the characteristics of the identified transfer function, there is $\Delta \tau_{12}=-60<0$ in the elements of the first row. Therefore, a time-delay compensation matrix $N_{\tau }(s)=diag(1,e^{-60s})$ is set. In the transfer function of the compensated system, there is $\Delta \tau_{21}=-30<0$ in the elements of the second row, and the elements in the first row have met the time-delay compensation conditions, and there is no redundancy for further compensation. Thus, an approximate transformation is carried out on the lag time of $g_{22}$ in the new transfer function:(13)$$g_{22}=\frac{-11.8e^{-90s}}{{\left(163s+1\right)}^{2}}\approx \frac{-11.8e^{-60s}}{{\left(163s+1\right)}^{2}(30s+1)}$$

After time-delay compensation, it is found that the relative order of the second-row elements in the transfer function matrix has $\Delta r_{21}=-1<0$. Therefore, relative compensation processing is carried out, and the step-down processing is carried out on the $g_{22}$ element:(14)$$g_{22}=\frac{-11.8e^{-60s}}{{\left(163s+1\right)}^{2}(30s+1)}\approx \frac{-11.8e^{-60s}}{\left(326s+1)(30s+1\right)}$$

Up to this point, there is no situation that requires compensation in the transfer function matrix, and the designed inverse decoupler is as follows:(15)$$Do(s)=\left[\begin{array}{cc} 0 & -\frac{3.3{\left(260s+1\right)}^{2}}{2.6{\left(150s+1\right)}^{2}}\\ \frac{5.6(326s+1)(30s+1)}{11.8{\left(180s+1\right)}^{2}} & 0 \end{array}\right]$$

### 4.1. Controller Parameter Setting

According to the fusion process of the control algorithm in Section 3.1, it can be inferred that the performance of the controller designed based on this algorithm will primarily be affected by the following six parameters: the prediction time domain P, control time domain M, continuous function discrete time $T_{s}$, weight coefficient between the output error of the regulation system and the control increment r, error weight coefficient $W_{1}$, and control weight coefficient $W_{2}$. Taking the main steam pressure in a combustion system as an example, the control variable method was used to analyze the key parameters that affect the performance of the controller. The initial error covariance matrix, process noise covariance, and measurement noise covariance in the ESKF-MPC algorithm were set as follows: $P_{0}={\left[\begin{array}{ccc} 1 & 0 & 0\\ \vdots & \ddots & \vdots \\ 0 & 0 & 1 \end{array}\right]}_{\left(n_{1}+m_{1})\times (n_{1}+m_{1}\right)}$, where $n_{1}$ represents the number of state quantities of the system and $m_{1}$ represents the number of output quantities of the system; $R={\left[\begin{array}{ccc} 1 & 0 & 0\\ \vdots & \ddots & \vdots \\ 0 & 0 & 1 \end{array}\right]}_{\left(n_{1}+m_{1})\times (n_{1}+m_{1}\right)}$; $Q=diag([\begin{array}{cc} 1 & 1 \end{array}])$. At 10 s, a step signal was first added to the main steam pressure control loop, and then a step signal was added to the bed temperature control loop after 6000 s. The total simulation time was set to 10,000 s.

(1)The influence of the prediction horizon P.

Based on the identified model’s dynamic response time scale, the prediction time-domain values of P were, respectively, taken as 20, 40, 60, and 80, the control time domain M was 2, the discrete time $T_{s}$ was 40, the adjustment weight r was 0.3, the error weight $W_{1}$ was [1; 1.5], and the control weight $W_{2}$ was [50; 40]. The output results of the main steam pressure in the system are shown in Figure 5. From Figure 5, it is evident that, with other parameters held constant, the prediction horizon significantly influenced the stability of the system. As the prediction horizon increased, the system stability improved notably, with the overshoot decreasing from 9.1% to 5.8%. When n became excessively large, the system’s dynamic performance changed minimally, but computational time increased significantly. Therefore, it was crucial to select an appropriate prediction horizon value based on the actual requirements of the CFB boiler system.

(2)The influence of the control horizon M.

When the control time domain M was taken as 2, 4, 6, and 8, the prediction time domain P was 40, the discrete time $T_{s}$ was 40, the adjustment weight r was 0.3, the error weight $W_{1}$ was [1; 1.5], and the control weight $W_{2}$ was [50; 40]. The output results of the main steam pressure prediction are shown in Figure 6. As can be seen from Figure 6, when M was 2, the system could track the set target smoothly, without vibration and with a small overshoot, only 6.4%, which was conducive to ensuring the stability of the control system. With the increasing of M, the rapidity of the system was improved, and the rise time was obviously shortened. However, it was accompanied by the large overshoot and oscillation of the system, and the maximum overshoot was 23.59%. Therefore, the controller parameters were adjusted to select a smaller control time-domain value on the basis of measuring the flexibility and stability of the system.

(3)The influence of the discrete time $T_{s}$.

When the discrete time $T_{s}$ was taken as 20, 40, and 60, respectively, the prediction time domain P was 40, the control time domain M was 2, the adjustment weight r was 0.3, the error weight $W_{1}$ was [1; 1.5], and the control weight $W_{2}$ was [50; 40]. The prediction outputs of the main steam pressure are shown in Figure 7. As can be seen from Figure 7, the control performance tended to be stable and smooth as the discrete time increased. A smaller discrete time not only brought a larger overshoot to the coupled primary wind fast response loop but also easily caused irregular fluctuations in the output. The discrete time was reasonably selected according to the expected control effect of the combined coupling loop of the combustion system.

(4)The influence of the adjustment weight r.

When the adjustment weight r was set to 0.1, 0.3, 0.5, and 0.7, respectively, the prediction time domain P was 40, the control time domain M was 2, the discrete time $T_{s}$ was 40, the error weight $W_{1}$ was [1; 1.5], and the control weight $W_{2}$ was [50; 40]. The output results of the main steam pressure prediction are shown in Figure 8. As can be seen from Figure 8, when r=0.1, the system responded fastest, but there was a large oscillation and steady-state deviation of the system, and the output curve continued to fluctuate and be unstable. When r=0.3, the overshoot of the system did not oscillate, and the overshoot was 6.4%. When r=0.5 and r=0.7, the overshoot of the system decreased further; the overshoot decreased to 4.5% and 3.6%, respectively, and the stability of the system was strengthened. From the formula derivation and the figure below, it can be seen that the adjustment weight had a great impact on the system control increment, and the stability of the system control could be improved by setting the adjustment weight reasonably.

(5)The influence of the error weight $W_{1}$.

When the error weight $W_{1}$ was set as [0.5; 1.5], [1; 1.5], and [1.5; 1.5], respectively, the prediction time domain P was 40, the control time domain M was 2, the discrete time $T_{s}$ was 40, the adjustment weight r was 0.3, and the control weight $W_{2}$ was [50; 40]. The output results of the main steam pressure prediction are shown in Figure 9. As can be seen from Figure 9, when the error weight coefficient affecting the main steam pressure loop was successively increased, the overshoot of the system increased from 4.0% to 6.4% and then to 8.5%, and the response speed of the system was significantly improved. When the weight coefficient of the main steam pressure loop increased from 0.5 to 1, the adjustment time of the system decreased from 1105 s to 1084 s. When the weight coefficient increased from 1 to 1.5, the adjustment time of the system continued to decrease from 1084 s to 1052 s. It can be seen that increasing the weight coefficient would make the system more sensitive to the output error, thus speeding up the response to the error. But it would also bring greater overshoot. For the selection of error weight coefficients, the requirements of control stability and tracking speed should be considered comprehensively.

(6)The influence of the control weight $W_{2}$.

When the control weight $W_{2}$ was taken as [30; 40], [40; 40], and [50; 40], respectively, the prediction time domain P was 40, the control time domain M was 2, the discrete time $T_{s}$ was 40, the adjustment weight r was 0.3, and the error weight $W_{1}$ was [1; 1.5]. The output results of the main steam pressure prediction are shown in Figure 10. As can be seen from Figure 10, when the control weight of the main steam pressure loop was 30, the overshoot of the system was 8.8%, and the adjustment time was 1043 s. When the value was 40, the overshoot of the system was 7.4%, and the adjustment time was 1069 s. If the value was 50, the system overshoot was 6.4%, and the adjustment time was 1083 s. When the control weight coefficient kept increasing, the overshoot amplitude of the system decreased, and the adjustment time increased slightly. Therefore, the appropriate value of the control weight coefficient was selected to maintain the system with a small overshoot and a fast response speed.

### 4.2. Tracking Performance Comparison

According to the combustion system model, four control strategies, ID-PI, ID-LADRC, MPC, and ESKF-MPC were set, respectively, for comparison. PI controller parameters were set according to Lambda tuning method, and the main steam control loop were set as $k_{p1}=0.0163$ and $k_{i1}=0.00027$; the bed temperature control loop values were set to $k_{p2}=-0.00355$ and $k_{i2}=-0.000094$. LADRC controller main steam control loop values were set to $k_{p,1}=0.008$, $\omega_{0,1}=0.08$, and $b_{0,1}=0.9$; the bed temperature control loop values were set to $k_{p,2}=0.008$, $\omega_{0,2}=0.08$, and $b_{0,2}=-2.4$. In the MPC controller, N=60, $N_{u}=2$, and $T_{s}=30$, the weight coefficient of the control variable was [0; 0], the weight coefficient of the output variable was [0.014; 0.012], and the upper and lower limits of the control variable were set to [−1, 1]. Based on the parameter adjustment rules derived in Section 4.1, the ESKF-MPC controller sets each parameter to P=60, M=2, $T_{s}=30$, r=0.2, $W_{1}=\left[0.05;0.055\right]$, and $W_{2}=\left[15;50\right]$. The simulation time was set to 6000 s, and the step with an amplitude of 1 was performed in the bed temperature control loop and the main steam pressure loop of the combustion system, respectively. This can be compared with the control effect of the four control strategies designed above.

Figure 11 and Figure 12 show that the ESKF-MPC control strategies all showed the fastest response speed in the step process, followed by MPC, ID-LADRC, and ID-PI. The performance indicators of the system tracking process under each control policy were as follows.

According to Table 1, compared with other control strategies, although ESKF-MPC control strategies had the largest overshoot, all overshoots in the statistical results were less than 2%. It could be basically considered that all response processes moved smoothly from one steady state to another without obvious overshoot phenomena. Comparing the adjustment time, it could be found that ESKF-MPC was the shortest among all control schemes, indicating that the control algorithm could make the control system have excellent tracking performance and could respond quickly to the change in excitation.

In addition, for a coupled combustion system, a single circuit had different response characteristics under different control strategies. For the PI and LADRC control strategies with an inverse decoupling strategy, there was a little fluctuation in the bed temperature loop during the main steam pressure step, while there was no fluctuation in the main steam pressure loop during the bed temperature loop step. For the slight fluctuation of the bed temperature loop in the main steam pressure step process, the reason was that the compensation process was processed by order reduction and approximation when designing the inverse decoupling controller, resulting in a slight deviation of the inverse decoupling model, and thus the decoupling effect cannot fully reach the ideal state. However, on the whole, the control scheme using the inverse decoupling strategy had the best decoupling effect. For MPC and ESKF-MPC control strategies, their decoupling abilities were due to the internal model predictive control algorithm, and the decoupling effect was lower than that of the inverse decoupling strategy. When the main steam pressure stepped, the overshoot of the bed temperature loop under the MPC strategy was 0.168, and the overshoot of the ESKF-MPC strategy was 0.139. When the bed temperature stepped, the overshoot of the MPC strategy was 0.115, and that of the ESKF-MPC strategy was 0.086. For the disturbance caused by system coupling, the ESKF-MPC control algorithm proposed in this paper could quickly minimize the influence of these disturbances by its disturbance estimation and compensation capabilities, so that the control performance index results could show a smaller overshoot amplitude.

### 4.3. Comparison of Anti-Interference Performance

In order to verify the immunity of the four control strategies in the combustion control system of the CFBB unit, a disturbance with an amplitude of 0.1 was applied at 8000 s of the combustion system after stabilization. The controller parameter settings in each control strategy were consistent with those in the previous section. The effects of different control strategies in the main steam pressure and bed temperature control circuits are shown in Figure 13 and Figure 14, respectively.

As can be seen from Figure 13, when the system applied interference in the main steam pressure loop at 8000 s, the overshoot of ID-PI was 20.46%, and the adjustment time was 3346 s. The overshoot of ID-LADRC was 19.17%, and the adjustment time was 2831 s. The overshoot of MPC was 12.43%, and the adjustment time was 1756 s. The overshoot of ESKF-MPC was 9.76%, and the adjustment time was 1179 s. The control strategies of the system’s coupled bed temperature control loops were as follows: the overshoot of MPC was 0.021, and the time interval to reach the initial steady state again was 1859 s. The overshoot of ESKF-MPC was 0.016, and the time interval to reach the initial steady state again was 858 s. The PI and LADRC circuits were less affected by coupling interference because of their inverse decoupling measures.

As can be seen from Figure 14, when the system applied interference in the bed temperature loop at 8000 s, the overshoot under the ID-PI control strategy was 93.25%, and the adjustment time was 2468 s. Under the ID-LADRC control strategy, the overshoot was 83.45%, with an adjustment time of 2046 s. Under the MPC control strategy, the overshoot was 55.88%, and the adjustment time was 1324 s. For the ESKF-MPC strategy, the overshoot was 43.16%, and the adjustment time was 666 s. The performance of the coupled main steam pressure circuit for each control strategy was as follows: the MPC overshoot was 8.92%, with a regulation time of 1371 s, while the ESKF-MPC overshoot was 7.97%, and the adjustment time was 855 s. Additionally, the PI and LADRC main steam pressure control circuits remained unaffected after the implementation of inverse decoupling.

### 4.4. Comparison of Robust Performance

The robustness of the combustion control system was verified by the Monte Carlo test. The gain coefficient and inertia time of the controlled object were perturbed by ±30% by this method. In the four control strategies, the ITAE, overshoot, and adjustment time were selected to repeat the test 200 times with the controller parameters unchanged.

As shown in Figure 15, it can be clearly observed from the three-dimensional figure that the comprehensive performance of different strategies was sequentially ranked as ESKF-MPC, MPC, ID-LADRC, and ID-PI. To more meticulously compare the advantages and disadvantages of various control strategies, two of the indicators were arbitrarily selected for graph drawing, respectively. Through comparison, it was found that the distribution span of the overshoot indicators of each control strategy was relatively large. The main reason was that the performance of the controller deteriorated after the model parameters were perturbed. The distribution statistics of the adjustment time and ITAE indicators are shown in Table 2.

From the table, it can be observed that the distribution of various indicators in the ESKF-MPC and MPC strategies was relatively concentrated, while the distribution of various indicators in the ID-LADRC and ID-PI strategies was relatively scattered. Among them, the indicators of the ID-PI strategy were the most scattered, and those of the ESKF-MPC strategy were the most concentrated. This indicates that the traditional PI controller had great difficulty in maintaining the stability of the system when the model of the controlled object underwent large-scale changes. In contrast, the MPC controller improved by integrating the ESKF had the strongest stability. ESK-MPC exhibited better dynamic response characteristics compared to LADRC, compensated by an ESO and MPC with a receding horizon optimization function. The powerful performance of the ESKF-MPC strategy benefited from the extended state Kalman filter. When model parameters changed, the Kalman gain coefficient was continuously updated through the estimated covariance results of this filter. Then, the impact of system biases was adaptively eliminated through the dynamic gain coefficient.

As shown in Figure 16, the control performance of different controllers in the bed temperature loop was basically similar to that of the main steam loop. The ESKF-MPC controller demonstrated the best overall performance, followed by the MPC controller, ID-LADRC controller, and ID-PI controller. The statistical results of the performance indicators of each controller in the Monte Carlo experiment are shown in Table 3.

As can be seen from Table 3, although the overshoot under the ESKF-MPC strategy was the largest among the four strategies, it was only 0.12% higher than that of the conventional PI strategy and 0.4% more than that of the LADRC strategy with the best overshoot performance. Basically, the overshoot performance under each strategy could be considered to be close. However, it showed significant advantages in terms of simulation time and comprehensive performance. This is also owed to the extended state Kalman filter. Using the algorithm of this filter, not only could the state variables of the changing system be accurately estimated but also the disturbances caused by the coupled main steam pressure loop could be accurately estimated. Subsequently, the prediction model was continuously corrected to ensure the accuracy of the model, laying an important foundation for the accurate control of the subsequent MPC. The dynamic gain adjustment mechanism in the filter reduced the sensitivity of the controlled system to system errors, minimized factors that could cause system overshoot, and thus effectively ensured the system’s rapid response and stable tracking.

### 4.5. Application in Actual Continuous Change Working Conditions

In order to further verify the effectiveness of the proposed control strategy, the strategy was applied to a 300 MW class CFBB under continuously changing combustion conditions, and the control effect was compared with the PI control strategy determined by the engineering setting method. The load conditions in the continuous change process of the unit are shown in Figure 17. First, the nonlinear model of the unit’s TITO combustion system was identified during this operation period. The fitting rates of the identified models were 74.88% for the main steam pressure and 80.71% for the bed temperature, respectively. The identified combustion system model was as follows:(16)$$\left[\begin{array}{l} P(s)\\ T(s) \end{array}\right]=\left[\begin{array}{cc} \frac{4.6923e^{-4}\cdot s-8.8269e^{-9}}{s^{2}+2.6340e^{-4}\cdot s+1.3781e^{-11}} & \frac{-1.5438e^{-4}\cdot s-1.2036e^{-9}}{s^{2}+1.0719e^{-4}\cdot s+2.0309e^{-8}}\\ \frac{-0.0017\cdot s+1.1909e^{-5}}{s^{2}+0.0081e^{-4}\cdot s+1.1724e^{-5}} & \frac{0.0019e^{-4}\cdot s-1.9020e^{-6}}{s^{2}+0.0297e^{-4}\cdot s+2.8951e^{-6}} \end{array}\right]\left[\begin{array}{l} M(s)\\ V(s) \end{array}\right]$$

Under the same control input, the dynamic response comparison results between the predicted output data of the identified model and the measured data are shown in Figure 18 and Figure 19. As shown in Figure 18 and Figure 19, whether the main steam pressure output or the bed temperature output of the identification model, the changing trend of the model’s predicted data was basically consistent with the measured data, so it can be considered that the identification model has high accuracy.

Then, ESKF-MPC and PI control simulations were built, respectively, in Matlab, to compare the control effects of main steam pressure and bed temperature under continuously changing working conditions. After tuning the ESKF-MPC controller parameters, the prediction time domain was 92, the control time domain was 6, the discrete time was 120, the adjustment weight was 0.6, the error weight was [0.38; 0.01], the control weight was [0.04; 0.1], and the frequency converter instruction constraints of each control quantity were 0~50 Hz. PI controller parameters were adjusted by the engineering setting method; the main steam pressure loop values were $k_{P1}=2.8$ and $k_{i1}=0.0025$; the bed temperature loop values were $k_{P2}=-0.12$ and $k_{i2}=-0.001$.

In the ESKF-MPC control strategy, the objective function was mainly the quadratic form of the output error and control quantity of steam pressure and bed temperature. Then, it was converted to the standard form for optimization control by using the quadprog optimization solver in Matlab. In addition, considering the complexity of bed temperature control, the set value of the bed temperature was not given in the actual control system, so the measured value of the bed temperature was set as the tracking target during the simulation strategy verification process. The control effects under each control strategy are shown in Figure 20 and Figure 21. As can be seen from Figure 20, the main steam pressure of the boiler under the ESKF-MPC control strategy could quickly and accurately track the set value without large overshoot and shock. However, under the PI strategy, the main steam pressure tracking value showed significant overshoot and fluctuation in the continuous variable condition tracking process or in the transition process from a steady-state condition to another steady-state condition, and the deviation could not be eliminated quickly to reach the set value. Figure 21 shows that the bed temperature tracking fluctuation of ESKF-MPC strategy was gentle, and the tracking performance was superior compared with the PI strategy. The reason was that ESKF-MPC compensated the model deviation in time through the extended state Kalman observer and used MPC control targeting to minimize the error of the system’s output and set point in the prediction time domain. According to the control performance of the whole combustion system, the stability of the main steam pressure and bed temperature had been significantly improved after the ESKF-MPC strategy was adopted, which provided a strong guarantee for the operation economy of the whole system and the operation reliability of the control actuator.

## 5. Conclusions

During the flexible operation of a CFBB, the instability of combustion inside the furnace leads to frequent fluctuations in signals such as the main steam pressure and the bed temperature. Moreover, the dynamic characteristics of the combustion system are constantly changing over time, making it difficult to establish an accurate model. In response to these issues, a multi-variable MPC control scheme that utilizes the ESKF to conduct real-time estimation and compensation for disturbance terms has been proposed. The control scheme can cope well with the problem of the deterioration of the control system performance caused by the surge of uncertain factors after the flexible operation of CFB units. The main conclusions are as follows:(1)For the multi-variable, strongly coupled, and nonlinear CFBB combustion system, the ESKF-MPC control strategy is well suited to handle this complex multi-input and multi-output control scenario. This strategy takes into account the interactions between system variables during the parameter setting process of the controller, thereby ensuring optimal overall control performance. For a system with such coupling, the control algorithm of PI and LADRC designed for single-loop systems must implement a decoupling strategy to achieve the desired control performance.(2)By expanding the “total disturbance” inside and outside the system into new observed variables, the ESKF can accurately estimate the values of all state variables and disturbances in the system at each moment. This approach effectively addresses the issue of low control performance caused by system model errors and overcomes the traditional MPC reliance on an accurate prediction model.(3)Compared to MPC control, ID-PI control, and ID-LADRC control with decoupling strategies, the proposed ESKF-MPC composite control method demonstrates the shortest adjustment time during the set point tracking process, with almost no overshoot. When disturbances are introduced to the two coupled control loops of the system, each loop exhibits minimal overshoot and the fastest recovery time. Additionally, when model parameters undergo changes, the ITAE index value remains relatively small under the ESKF-MPC strategy.(4)Under continuously changing operating conditions, the main steam pressure loop of the CFBB combustion system using the ESKF-MPC strategy achieves rapid tracking and disturbance elimination, with no significant overshoot or output shock. This effectively prevents operational instability caused by frequent fluctuations in the main steam pressure. In the coupled bed temperature loop, the ESKF-MPC strategy significantly reduces the bed temperature fluctuation range compared to the PI strategy, which helps to ensure the economic operation of the unit.

In summary, the control scheme based on ESKF-optimized MPC proposed in this paper improves the model-based optimal control scheme to a model-free one, greatly enhancing the robustness and anti-interference performance of MPC in the face of complex and changeable controlled systems. Although its good performance has been evaluated in the current simulation experiments, there are still several issues that require further research. (1) For the combustion system of circulating fluidized bed boilers participating in flexible operation, develop an external system or configuration logic suitable for engineering applications. (2) When the dynamic characteristics of the system change drastically, the proposed control optimization scheme is insufficient to guarantee performance advantages. We should address the problem of how this can be further optimized to achieve adaptive control. (3) In engineering applications, develop more efficient solution algorithms to reduce the online computational burden of MPC.

## Figures and Tables

**Figure 1 sensors-25-01262-f001:**
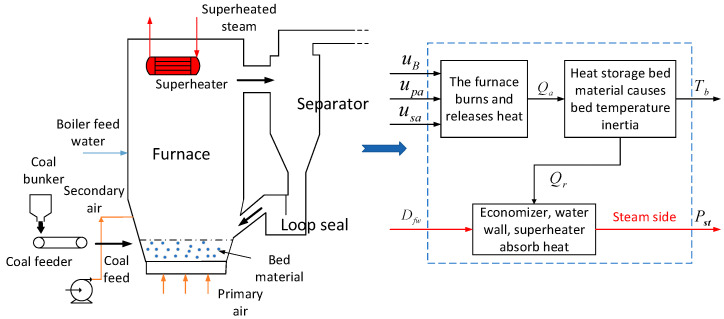
Simplified schematic diagram of CFBB combustion process.

**Figure 2 sensors-25-01262-f002:**
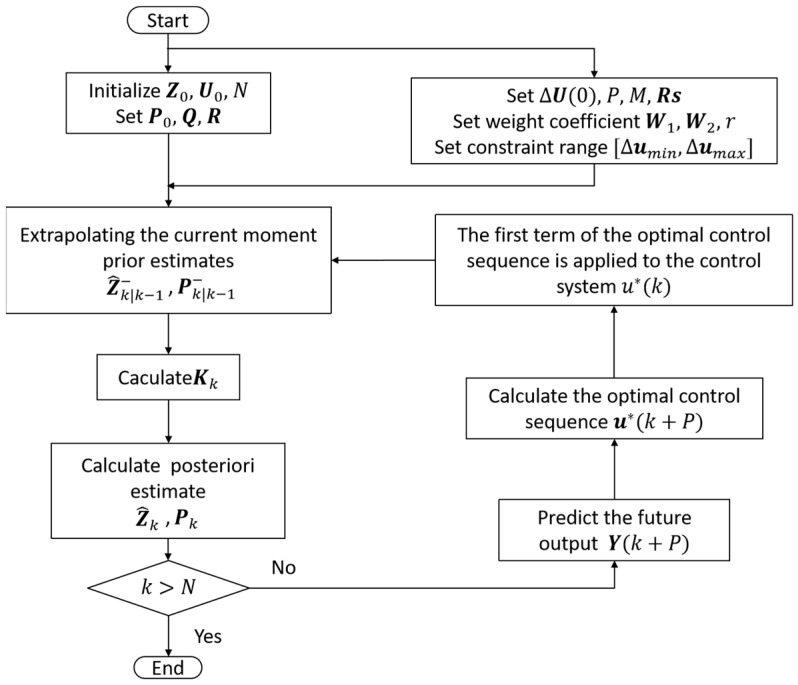
MPC algorithm flow of ESKF fusion.

**Figure 3 sensors-25-01262-f003:**
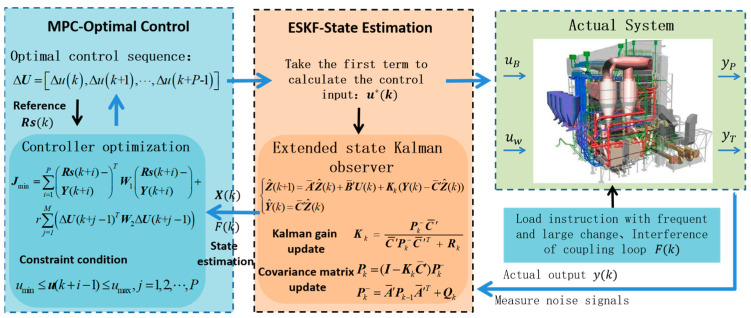
This ESKF-MPC control architecture diagram of the CFBB combustion system.

**Figure 4 sensors-25-01262-f004:**
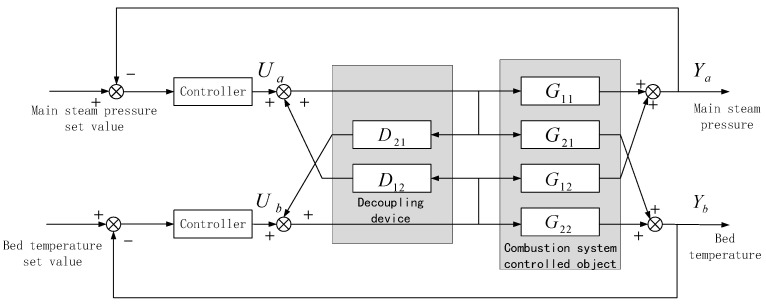
Inverted decoupling control block diagram of CFBB combustion system.

**Figure 5 sensors-25-01262-f005:**
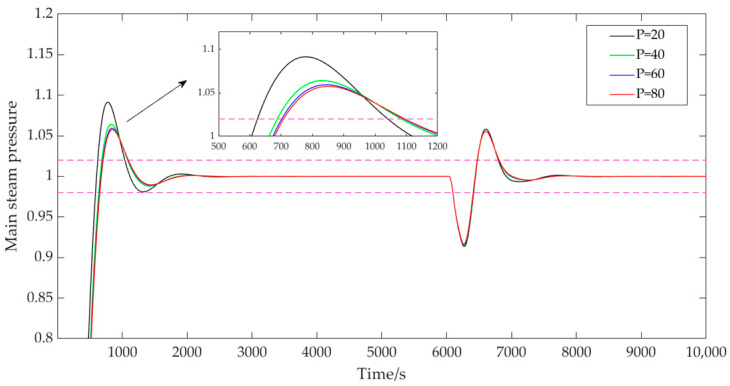
System output with variable prediction time domains.

**Figure 6 sensors-25-01262-f006:**
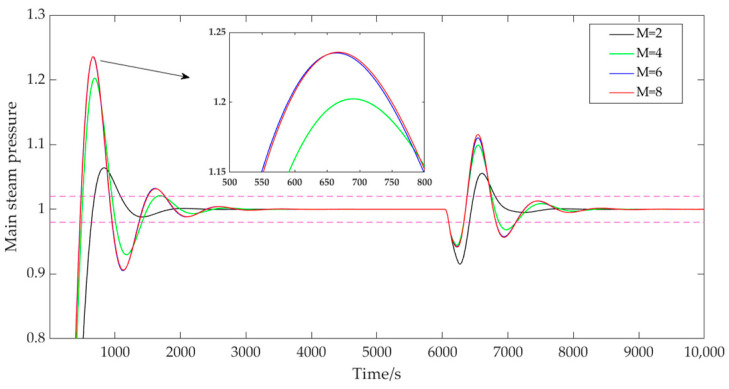
System output with variable control time domains.

**Figure 7 sensors-25-01262-f007:**
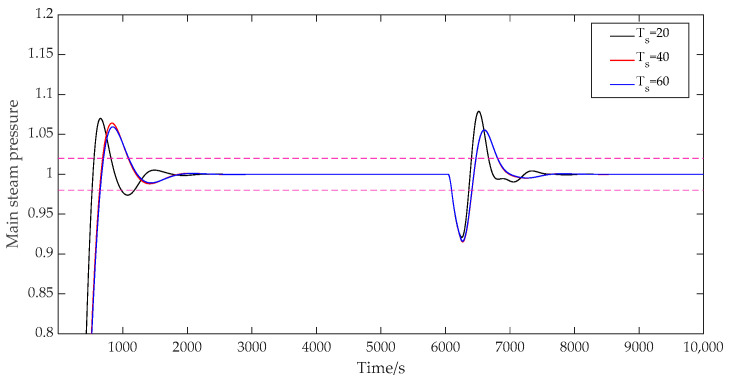
System output with variable discrete times.

**Figure 8 sensors-25-01262-f008:**
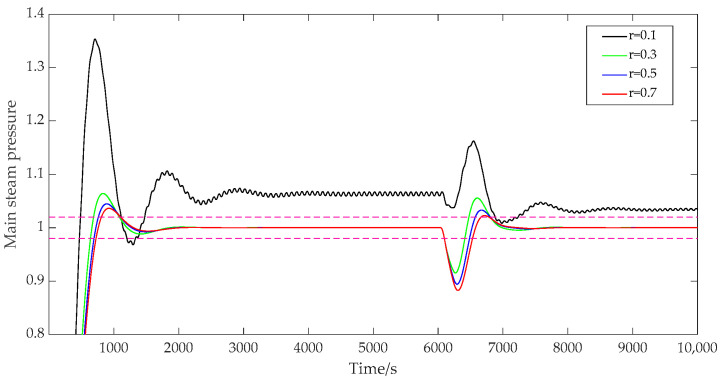
System output with variable adjusting weights.

**Figure 9 sensors-25-01262-f009:**
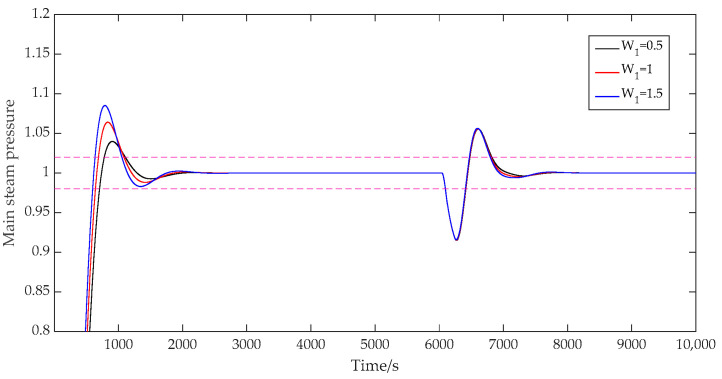
System output with variable error weights.

**Figure 10 sensors-25-01262-f010:**
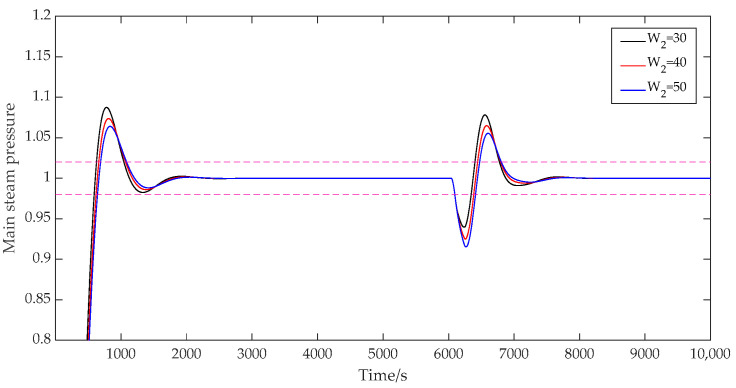
System output with variable control weights.

**Figure 11 sensors-25-01262-f011:**
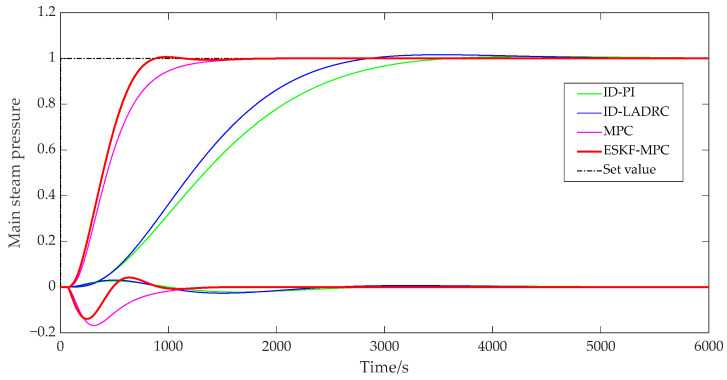
Step response curve of main steam pressure.

**Figure 12 sensors-25-01262-f012:**
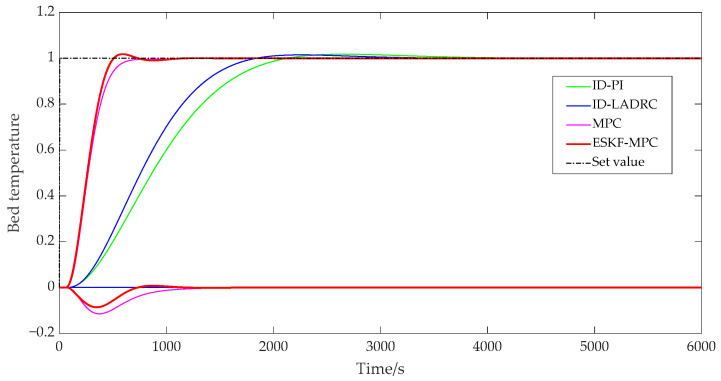
Step response curve of bed temperature.

**Figure 13 sensors-25-01262-f013:**
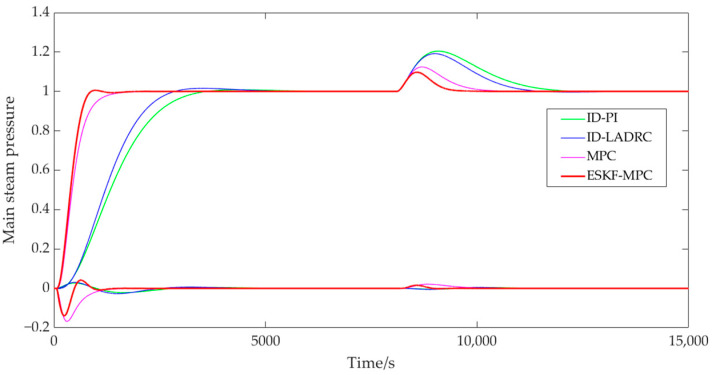
Main steam pressure disturbance process curve.

**Figure 14 sensors-25-01262-f014:**
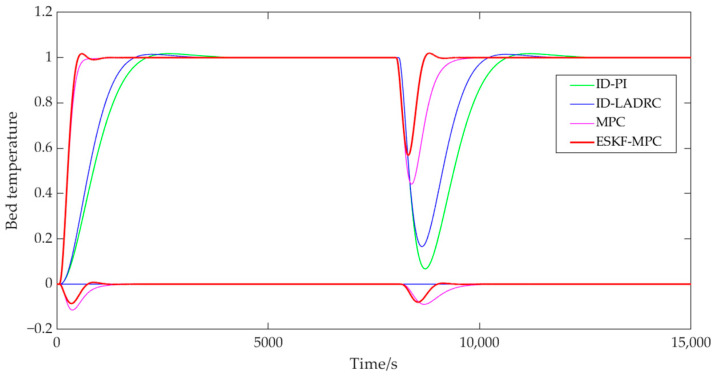
Bed temperature disturbance process curve.

**Figure 15 sensors-25-01262-f015:**
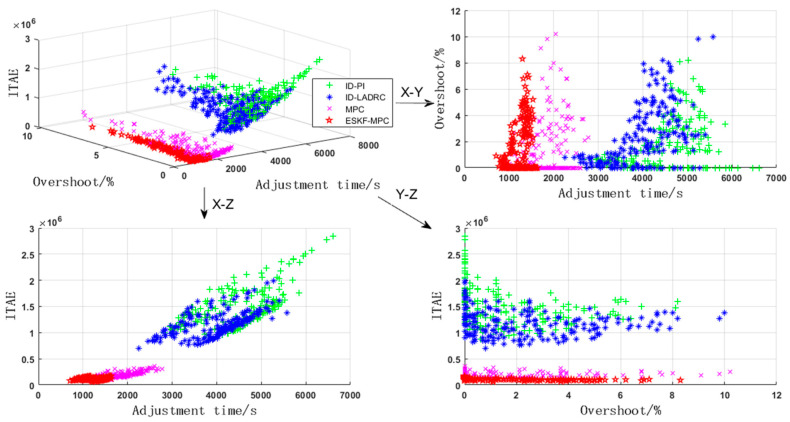
Monte Carlo test results of the main steam pressure circuit.

**Figure 16 sensors-25-01262-f016:**
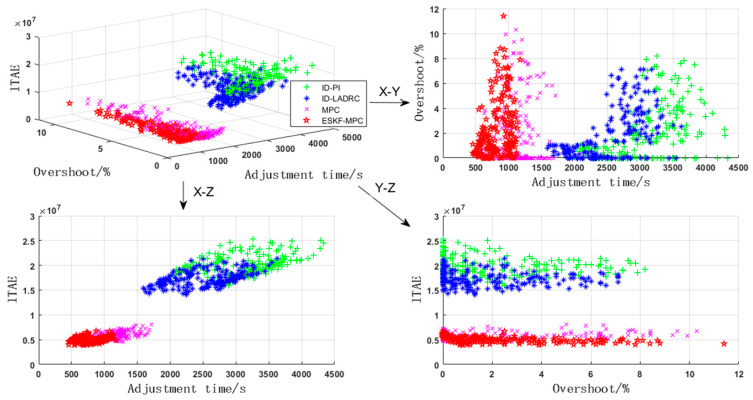
Monte Carlo test results of bed temperature circuit.

**Figure 17 sensors-25-01262-f017:**
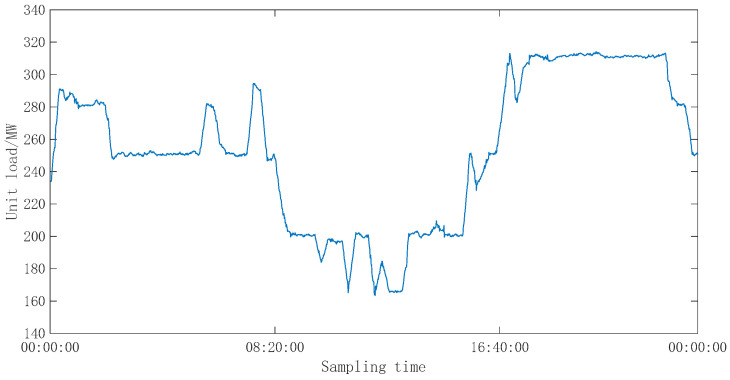
Operating curve of the unit under working conditions.

**Figure 18 sensors-25-01262-f018:**
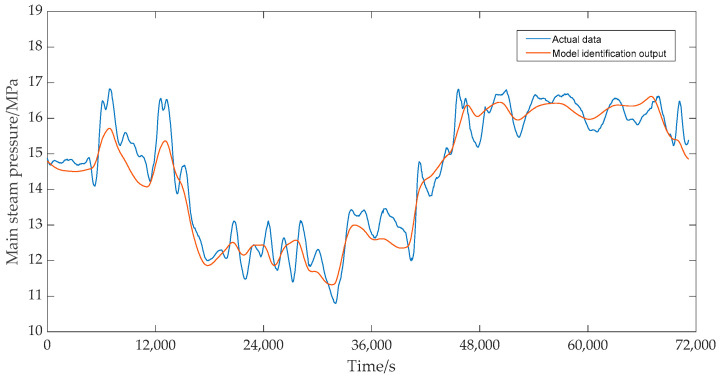
Main steam pressure test results.

**Figure 19 sensors-25-01262-f019:**
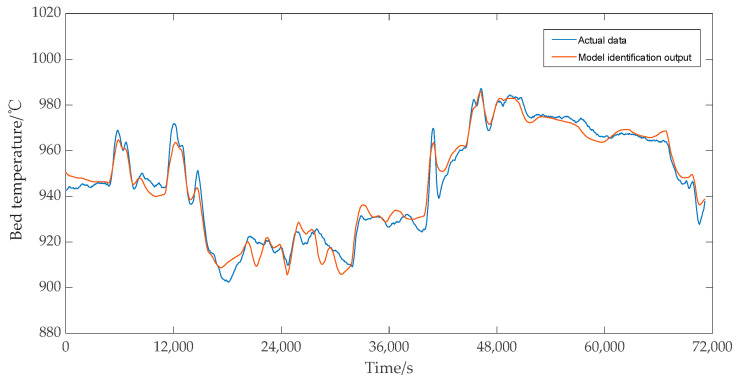
Bed temperature test results.

**Figure 20 sensors-25-01262-f020:**
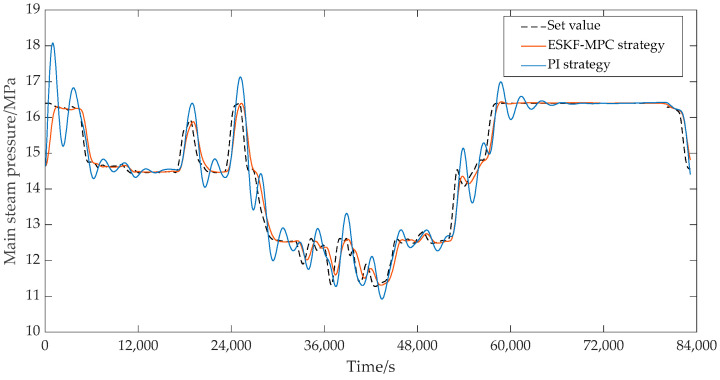
Effect of main steam pressure control circuit.

**Figure 21 sensors-25-01262-f021:**
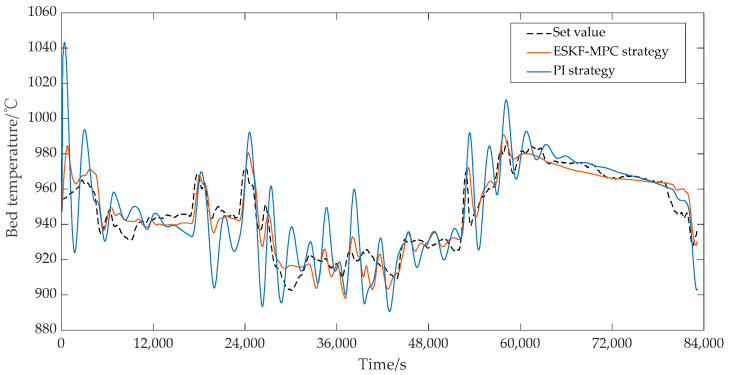
Effect of bed temperature control circuit.

**Table 1 sensors-25-01262-t001:** The performance indicators of the tracking process under four control strategies.

Control Strategy	Main Steam Pressure Loop		Bed Temperature Loop	
	Adjustment Time/s	Overshoot/%	Adjustment Time/s	Overshoot/%
ID-PI	3163	0.83	1949	1.75
ID-LADRC	2611	1.61	1675	1.47
MPC	1268	0	592	0
ESKF-MPC	794	0.63	476	1.81

**Table 2 sensors-25-01262-t002:** Distribution of adjustment time and ITAE indicators for four control strategies.

Control Strategy	Adjustment Time		ITAE Range
	Distribution Range of Adjustment Time	Time Span	
ID-PI	2900~6000 s	3100 s	[0.87 × 10^6^, 2.84 × 10^6^]
ID-LADRC	2600~5000 s	2400 s	[0.7 × 10^6^, 1.99 × 10^6^]
MPC	1000~2600 s	1600 s	[1.02 × 10^5^, 3.56 × 10^5^]
ESKF-MPC	770~1600 s	830 s	[0.615 × 10^5^, 1.74 × 10^5^]

**Table 3 sensors-25-01262-t003:** Performance indicators of four control strategies.

Control Strategy	Distribution Range of Adjustment Time	ITAE Range	Mean Value of Overshoot
ID-PI	2000~4000 s	[1.59 × 10^7^, 2.54 × 10^7^]	2.13%
ID-LADRC	1590~3500 s	[1.4 × 10^7^, 2.14 × 10^7^]	1.85%
MPC	520~1600 s	[4.44 × 10^6^, 8.11 × 10^6^]	2.08%
ESKF-MPC	460~1100 s	[3.9 × 10^6^, 6.75 × 10^6^]	2.25%

## Data Availability

Data are contained within the article.

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
