# Peer review of "Flexible Optimal Control of the CFBB Combustion System Based on ESKF and MPC"

_sensors, 2025, doi:10.3390/s25041262_

Round 1

Reviewer 1 Report

Comments and Suggestions for Authors

A Model Predictive Control scheme based on the Extended State Kalman Filter is proposed in this manuscript to enhance the control of the CFBB combustion system. The writing is standard, and the conclusions are clear. It is recommended for publication after revision. Some suggestions:

1. Formula 9 is unreadable due to garbled text when converted to PDF.

2. Symbols in the figures should be consistent with those in the formulas.

3. Please verify the direction of the arrows in Figure 3.

4. Is the model in Formula 11 identified by the author or from other literature? If it is from other literature, a citation should be added.

5. The derivation of Formula 13 is unclear; please review it.

6. Section 4.1 discusses the impact of control parameters and contains many repetitions. Combining redundant expressions would be more conducive to readability.

7. The relationship between the parameter selection in Section 4.2 and Section 4.1 needs to be clarified.

8. In the curves of Section 4.2,  legends for the disturbance curve shoud be added.

9. The comparative conclusions in Section 4.4 would be clearer in a list format.

10. In Section 4.5, is the identification data derived from an actual boiler or a simulation model? The model identification results and the decoupling matrix can be provided.

11. The English language of the article needs improvement.

12. The role of the Extended State Kalman Filter can be analyzed in more detail during the simulation.

Comments on the Quality of English Language

It is recommended that the English expressions be carefully polished.

Reviewer 2 Report

Comments and Suggestions for Authors

Review of the draft manuscript ID sensors-3400267

Flexible Optimal Control of the CFBB Combustion System  Based on ESKF and MPC by Lei Han, Lingmei Wang, Enlong Meng ,Yushan Liu  and Shaoping Yin

The title of the draft manuscript is informative about the research paper. The research question is almost well justified given what is already known about the topic. The obtained results are presented in almost an appropriate way. Some figures need improvements in order to will be more readable. The depicted results are placed into context without being over interpreted. For more contribution, the Authors should compare their results with those in published works of other researchers. In general, the whole manuscript is well organized in chapters. Furthermore, the current work is well documented in the research field 30 references).  Major points in the article which needs clarification, refinement and/or additional information and suggestions are listed below:

1/ The Abstract does not contain specific information about the obtained results. So also, the abstract reads very general. Please include some quantitative data related to research outcomes. Moreover, the Abstract should explicitly list the elements of the work which are new and indicate why the manuscript should be published.

2/ In keywords, you should not give shortcuts, which are defined in the content of the draft manuscript.

3/ Authors should void lumped citations in the body text of the draft manuscript ID sensors-3400267. Please describe the referred works with at least two sentences.

4/ References should be cited consistently, i.e. with or without names of authors in the draft manuscript.

5/ The main aim of the research paper should be clearly defined in the introduction section.

6/ The authors should precisely identify and explicitly articulate their own concrete achievements. In the current version of the draft manuscript, their own contribution of a scientific nature is not obvious.

7/ Lacking of primary air in the diagram CFB combustion system, which is depicted in Fig. 1.

8/ It would be better if all symbols used in the body text of the draft manuscript were explained in a separate section called Nomenclature or Notation.

9/ Figure 3 needs improvements because it is eligible. For more readability, authors should consider increasing the font size for the figure's descriptions.

10/ Why did you choose the specific input data for this investigation? Any advantages or limitations?

11/ It is a wrong practice to use the equations in the body text of the manuscript, without mentioning the source (literature data).

12/ Structure of the draft manuscript is almost writing optimal. Nevertheless, what is a practically meaning and usefulness of the obtained results? Provide some enhanced scope, beneficiaries, and benefits from your study.

13/ The validation of predicted data should be presented/highlighted properly.

14/ What do you mean by dashed red lines in Figure 7? The same question applies to figures 8, 9 and 10.

15/ The obtained results are not compared with published data by other researchers. For more contribution, the Authors should compare their results with those in relevant published works of other researchers.

16/ The graphs depicted in Figure 15 are illegible. For more readability, authors should consider resizing each graph. The same problem goes for figure 16.

17/ The reviewer suggests that future research and lines of research should be indicated listed and also highlighted at the end of the Conclusions section.

Authors should consider above-mentioned remarks in order to revise the manuscript. Reviewer thinks that a publication of the draft manuscript may be possible after a major revision.

Round 2

Reviewer 2 Report

Comments and Suggestions for Authors

TThe new version of the manuscript is better, and the authors have addressed my queries.